# MambaLLIE: Implicit Retinex-Aware Low Light Enhancement with Global-then-Local State Space

**Jiangwei Weng[1], Zhiqiang Yan[1], Ying Tai[2], Jianjun Qian[1], Jian Yang[1], Jun Li[1]***

[1]School of Computer Science and Engineering,
Nanjing University of Science and Technology, Nanjing, 210094, China
[2]School of Intelligence Science and Technology, Nanjing University, Suzhou, 215163, China
`{wengjiangwei,yanzq,csjqian,csjyang,junli}@njust.edu.cn`     `yingtai@nju.edu.cn`

## Abstract

Recent advances in low light image enhancement have been dominated by Retinex-based learning framework, leveraging convolutional neural networks (CNNs) and Transformers. However, the vanilla Retinex theory primarily addresses global illumination degradation and neglects local issues such as noise and blur in dark conditions. Moreover, CNNs and Transformers struggle to capture global degradation due to their limited receptive fields. While state space models (SSMs) have shown promise in the long-sequence modeling, they face challenges in combining local invariants and global context in visual data. In this paper, we introduce MambaLLIE, an implicit Retinex-aware low light enhancer featuring a global-then-local state space design. We first propose a Local-Enhanced State Space Module (LESSM) that incorporates an augmented local bias within a 2D selective scan mechanism, enhancing the original SSMs by preserving local 2D dependencies. Additionally, an Implicit Retinex-aware Selective Kernel module (IRSK) dynamically selects features using spatially-varying operations, adapting to varying inputs through an adaptive kernel selection process. Our Global-then-Local State Space Block (GLSSB) integrates LESSM and IRSK with layer normalization (LN) as its core. This design enables MambaLLIE to achieve comprehensive global long-range modeling and flexible local feature aggregation. Extensive experiments demonstrate that MambaLLIE significantly outperforms state-of-the-art CNN and Transformer-based methods. Our code is available at Project Page.

## 1 Introduction

Low light image enhancement is a challenging task in computer vision due to insufficient lighting and sensor degradation. Consequently, images often suffer from poor global visibility and local issues such as color distortion and noise. These degraded images can adversely affect human perception and related vision tasks, such as object detection [5] and depth estimation [57].

Traditional techniques, such as histogram equalization [1] and gamma correction [7], enhance images through global mapping operations. However, these global operations often struggle to address local degradation effectively. In recent years, many methods based on CNNs and Transformers have gradually come to dominate this field [53, 64, 15, 38, 56, 3]. CNN-based methods [53, 64, 15, 38, 55] have achieved significant advancements by effectively aggregating local information, thus substantially improving performance in low light enhancement. Nevertheless, the limited receptive field and weight-sharing strategy of CNNs result in a local reductive bias, making the models less adaptive to varying inputs. On the other hand, Transformer-based methods [56, 3, 62] achieve a

---

*Corresponding author

38th Conference on Neural Information Processing Systems (NeurIPS 2024).

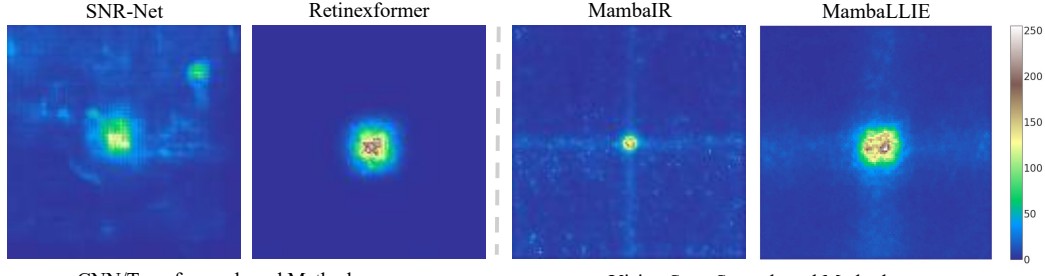

| SNR-Net | Retinexformer | MambaIR | MambaLLIE |
| CNN/Transformer based Methods | | Vision State Space based Methods | |

Figure 1: The Effective Receptive Field (ERF) visualization [34] for SNR-Net [56], Retinexformer [3], MambaIR [16] and our MambaLLIE. A broader distribution of bright areas signifies a larger ERF. The receptive field of SNR-Net is large but messy, due to the SNR-aware mechanism, Retinexformer achieves a larger receptive field of the central point, and MambaIR has the global receptive field, but presents the limited local perception. Only our proposed MambaLLIE achieves a global perception ability outwards from central point and preserves the large local receptive field.

larger and adaptive receptive field by emphasizing long-term dependencies through the self-attention mechanism. However, the vanilla attention mechanism scales quadratically with input size, resulting in significant computational overhead.

Recently, State Space Models (SSMs) [10, 31, 28] have garnered significant attention in the field of computer vision. These internal state space models demonstrate great potential for global information modeling with linear complexity. However, a straightforward implementation of vision state space models for low light image enhancement is inadequate. This is because SSMs are primarily designed for long-range modeling and lack the flexibility needed to capture local information effectively [66]. For instance, as illustrated in Fig. 1, the receptive field of MambaIR [16], a simple yet effective vision state space model, achieves longer-range dependencies compared to CNN and Transformer-based methods. Nevertheless, it falls short in refining local interactions.

In this work, we introduce MambaLLIE, a novel framework for enhancing low light images that integrates an implicit Retinex-aware approach within a global-then-local state space model. MambaLLIE not only explores the capabilities of state space models in low light image enhancement but also incorporates a Retinex-aware structure providing both explicit and implicit guidance. Our framework introduces a unique global-then-local state space block, enhancing global long-range degradation modeling and local feature aggregation through an augmented state space. Additionally, we incorporate a Retinex-aware selective kernel mechanism in the enhancement process, enabling adaptive modulation of illumination strength through specific spatial operations.

Our contributions and main findings can be summarized as follows: 1) We introduce a novel global-then-local state space block that integrates a local-enhanced state space module and an implicit Retinex-aware selective kernel module. This design effectively captures intricate global and local dependencies. 2) We devise an implicit Retinex-aware selective kernel mechanism to guide deeper neural representations, eliminating the need for complex structural design and constraints to estimate physical priors, the prior feature tends to segregate into independent positive and negative illumination components before integrating them, a capability lacking in explicit methods. 3) Experimental results on benchmark datasets and real-world evaluations consistently demonstrate the superior performance of our proposed method compared to state-of-the-art approaches.

## 2 Related work

**Low Light Image Enhancement.** Nowadays, the existing deep learning-based methods have mainly been categorized into end-to-end and Retinex-based methods [27]. To the best our knowledge, LLNet [33] firstly introduced a deep neural network for low light image enhancement by supervised learning. LightenNet [2] adopted the CNN for single image contrast enhancement. MBLLEN [35] proposed the multi-branch fusion within CNNs to extract rich features. Besides, SNR-Net [56], Restormer [62], LLFormer [22] and SCENet [36] adopted the self-attention mechanism to achieve excellent performance. However, all these end-to-end models mainly depend on the distribution of training

dataset and ignore the inherent illumination prior. As contrast, ZeroDCE [15], RUAS [30] and subsequent works [38, 9, 51] represent impressive solutions for image enhancement, as ones precisely using physical priors to enhance the images. However, due to the absence of an ideal reference for guidance, these methods usually exhibit a certain gap compared to supervised learning models.

As for supervised Retinex-based models, these methods aim to decompose the image into illumination and reflectance maps, and then enhance the image by optimizing these maps. For instance, Retinex-Net [53] divided image enhancement into decomposition, adjustment and reconstruction stages, which providing a good representation of image enhancement process. KinD [64] and URetinex-Net [55] further introduced the novel multi-branch and multi-stage frameworks, respectively. However, striking a balance between complexity and efficiency remains challenging for these methods. Recently, Retinexformer [3] simplified a one-stage Retinex-based low light enhancer with a efficient Transformer. Diff-Retinex [61] designed a transformer-based decomposition network and adopted generative diffusion networks to reconstruct the results. Overall, they typically applied the Retinex theory in a direct way, which may be limited for low light enhancement problem.

**Vision State Space Model.** State Space Model (SSMs) [11, 12, 13] are emerging new sequence models for deep learning, which first swept the natural language processing (NLP) community such as language understanding [42], content-based reasoning [66]. Recently, SSMs have also garnered considerable attention in computer vision (CV) tasks. To our knowledge, S4ND [39] first explored state space mechanism into CV tasks by swapping Conv2D and self-attention layers with S4ND layers in existing models. VMamba [31] bridged the gap between ordered sequences and non-causal visual images, enabling the extension of vision selective state space model with global receptive fields. Vim [65] proposed the bidirectional state space modeling with positional awareness, achieved the global visual perception. Furthermore, LocalMamba [18] was focused on the local scanning strategy, preservation of local context dependencies. EfficientVMamba [41] designed a light-weight SSMs with an additional convolution branch to learn both global and local representational features. MambaIR [16] employed convolution and channel attention to enhance the capabilities of the Mamba. But existing vision state space model do not pay enough attention on capturing local information, as vanilla SSMs are designed for long sequence and the invariant of local vision data is not taken into account in the existing vision state space models.

## 3 Methodology

This work aims to introduce a novel implicit Retinex-aware low light enhancer with global-then-local state space. In this section, we revisit the Retinex theory and the state space model, offering a concise overview. Following that, the details of our proposed MambaLLIE are introduced.

### 3.1 Preliminaries

**Retinex Theory.** The ideal Retinex theory [25] for low light enhancement assumes that the captured images can be decomposed into reflectance and illumination maps. Following [38, 45], explicit Retinex-based methods emphasize estimating either an illumination map while regarding the reflectance map as the enhanced result, or estimating concrete reflectance and illumination maps and then restoring the well-exposed images. Specifically, given a low light image $\mathbf{L} \in \mathbb{R}^{H \times W \times 3}$, where $H$ and $W$ represent height and width respectively, the derived maps can be denoted as:

$$\mathbf{L} = \mathbf{R} \cdot \mathbf{I}, \quad \mathbf{N} = \mathbf{L} \Big/ \tilde{\mathbf{I}}, \quad \mathbf{N} = \tilde{\mathbf{R}} \cdot \tilde{\mathbf{I}}, \tag{1}$$

where $\cdot$ denotes the element-wise multiplication, $\mathbf{R} \in \mathbb{R}^{H \times W \times 3}$ denotes reflectance map, a static property of captured objects; $\mathbf{I} \in \mathbb{R}^{H \times W}$ denotes illumination map; $\mathbf{N} \in \mathbb{R}^{H \times W \times 3}$ denotes enhanced images; $\tilde{\mathbf{R}}, \tilde{\mathbf{I}} \in \mathbb{R}^{H \times W \times 3}$ denotes the estimated reflectance and illumination maps, respectively.

Consequently, the former assumption $\mathbf{N} = \mathbf{L}/\tilde{\mathbf{I}}$ disregards the noise and artifacts resulting from sensor degradation in the captured images, rendering pixel-wise illumination adjustments inadequate. The latter $\mathbf{N} = \tilde{\mathbf{R}} \cdot \tilde{\mathbf{I}}$ aims to restore the reflectance and illumination maps to enhance the images. However, this requires the design of multiple branches and constraints to guide the training [64].

**State Space Model.** The SSMs, such as structured state space sequence models (S4) [12] and Mamba [10], can be regarded as the continuous linear time-invariant (LTI) systems [54]. Given an

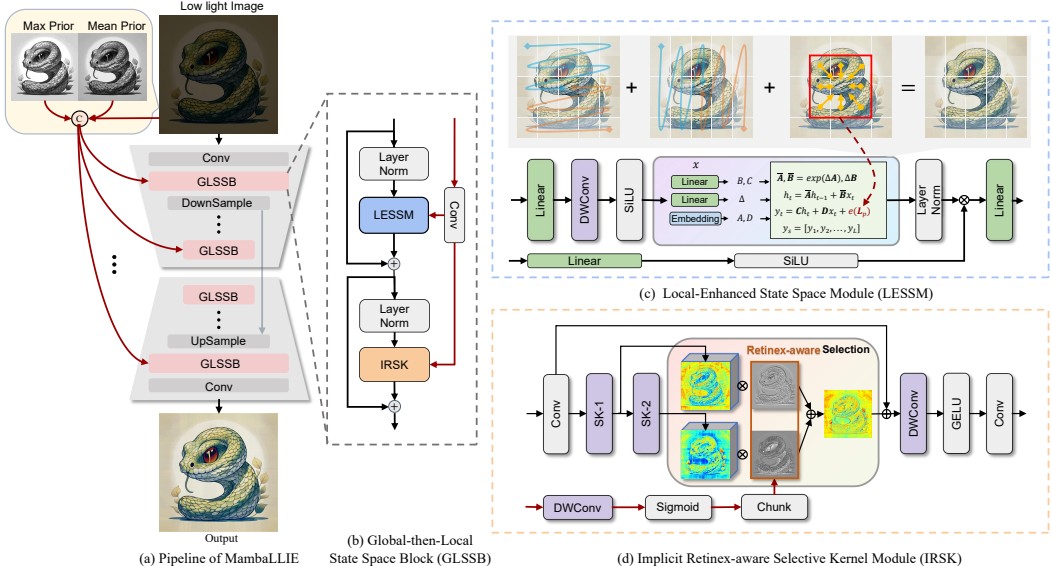

Figure 2: The overall pipeline of the proposed MambaLLIE. Our Global-then-Local State Space Block (GLSSB) integrates Local-enhanced state space module (LESSM) and implicit Retinex-aware selective kernel module (IRSK) with layer normalization as its core, where the maximum and mean maps of low light images can be regarded as a rough illumination prior of GLSSB. Besides, The local-enhanced design essentially introduces the local invariance into state space model, which can integrate the existing directional scan with our local-enhanced term into state space.

one-dimension sequence $x(t) \in \mathbb{R}$, it projects into a new one-dimension sequence $y(t) \in \mathbb{R}$ through the hidden state $h(t) \in \mathbb{R}^m$, the system can be defined as a linear ordinary differential equation:

$$
\begin{aligned}
h'(t) &= \mathbf{A}h(t) + \mathbf{B}x(t), \\
y(t) &= \mathbf{C}h(t) + \mathbf{D}x(t),
\end{aligned}
\tag{2}
$$

where $m$ denotes the state size, $\mathbf{A} \in \mathbb{R}^{m \times m}$, $\mathbf{B} \in \mathbb{R}^{m \times 1}$, $\mathbf{C} \in \mathbb{R}^{1 \times m}$ and $\mathbf{D} \in \mathbb{R}$ denotes state, input projection, output projection parameters, and kernel selective parameters.

As the raw state-space models are continuous, the systems adopt the discrete versions before feeding the computer, in which the zero-order hold (ZOH) is used to transform the continuous parameters $\mathbf{A}$ and $\mathbf{B}$ to discrete parameters $\bar{\mathbf{A}}$ and $\bar{\mathbf{B}}$ as follows

$$
\bar{\mathbf{A}} = \exp(\Delta \mathbf{A}), \quad \bar{\mathbf{B}} = (\Delta \mathbf{A})^{-1} (\exp(\Delta \mathbf{A}) - \mathbf{I}) \cdot \Delta \mathbf{B},
\tag{3}
$$

where $\Delta$ denotes the step size. Overall, the discretized version can be rewritten as:

$$
h_t = \bar{\mathbf{A}}h_{t-1} + \bar{\mathbf{B}}x_t, \quad y_t = \mathbf{C}h_t + \mathbf{D}x_t.
\tag{4}
$$

However, the current system remains static for varying inputs. To address this limitation, Mamba [10] introduces selective state space models, allowing parameters to adapt with the input, thereby enhancing selective information processing across sequences. This parameter selection mechanism can be expressed as:

$$
\bar{\mathbf{B}} = f_{\mathbf{B}}(x_t), \quad \bar{\mathbf{C}} = f_{\mathbf{C}}(x_t), \quad \Delta = \vartheta_{\mathbf{A}} (\mathbf{P} + f_{\mathbf{A}}(x_t)),
\tag{5}
$$

where $f_{\mathbf{B}}(x_t)$, $f_{\mathbf{C}}(x_t)$ and $f_{\mathbf{A}}(x_t)$ are linear functions that broadens feature to the hidden state dimensions. As SSMs are tailored for long sequences, it is limited in capturing complicated local information. As for visual data, VMamba [31], Vim [65], *etc.*, proposed the specific location-aware scan strategies to maintains the integrity of 2D image structures. However, the specific directed sequences overlook the vision information of pixels neighborhood structure. Inspired by [66], we explore a global-then-local state space, which receives the global perception before the details, supplementing the lack of local information.

## 3.2 Overall Pipeline

We first present the overall pipeline of our MambaLLIE, an U-shaped architecture as shown in Fig. 2(a), which includes encoding and decoding parts with the convolutional downsampling and upsampling layers. The encoder features are concatenated with the decoder features via skip connections. Next, We propose a global-then-local state space block (GLSSB) as the basic core of MambaLLIE, the max and mean priors concatenated with low light image are projected into GLSSB by convolutional layers. Therein, GLSSB is composed of the local-enhanced state space module (LESSM) and the implicit Retinex-aware selective kernel module (IRSK), interleaved with layer normalization.

**Illumination Prior.** Given a low light image $\mathbf{L} \in \mathbb{R}^{H \times W \times 3}$, we employ a $3 \times 3$ convolution layer to project the neural features $\mathbf{F} \in \mathbb{R}^{H \times W \times C}$ from input feature space, and then project features into each GLSSB, which will be described in Section 3.3. Besides, IRSK integrates original input, maximum prior $\mathbf{L}_{\max} \in \mathbb{R}^{H \times W}$ and mean prior $\mathbf{L}_{\mathbf{mean}} \in \mathbb{R}^{H \times W}$ as illumination prior $\mathbf{L}_{\mathbf{p}} \in \mathbb{R}^{H \times W \times 5}$,

$$\mathbf{L}_p = \text{Concat}\left(\mathbf{L}, \text{mean}(\mathbf{L}), \max(\mathbf{L})\right). \tag{6}$$

We first define $\mathbf{F}_g$ is the output of GLSSB. Subsequently, the downsampling layer and following GLSSB achieve the feature extraction to acquire the deep feature, which can be denoted as $\mathbf{F}_g \in \mathbb{R}^{\frac{H}{2^i} \times \frac{W}{2^i} \times 2^i C}$, where $i = 0, 1, 2$. Moreover, the feature is later concatenated with the upsampling layer with a symmetrical structure. Finally, using a $3 \times 3$ convolution layer projects into $\mathbf{F}_{\mathbf{out}} \in \mathbb{R}^{H \times W \times 3}$ and the enhanced image can be expressed as $\mathbf{N} = \mathbf{F}_{out} + \mathbf{L}$.

## 3.3 Global-then-Local State Space Block

As illustrated in Fig. 2(b), GLSSB follows the LayerNorm, LESSM, LayerNorm and IRSK flow, motivated by Transformer [46] and Mamba[10] usage of similar structures in a basic block. Given the input feature, it first undergoes the LayerNorm and LESSM to capture the local-enhanced global information. the above process can be denoted as:

$$\mathbf{M} = \text{LESSM}\left(\text{LN}\left(\mathbf{F}_g^{i-1}\right)\right) + \mathbf{F}_g^{i-1}. \tag{7}$$

And then, another LN and our proposed IRSK are used for Retinex-aware guidance. The above process can be formulated as:

$$\mathbf{F}_g^i = \text{IRSK}\left(\text{LN}\left(\mathbf{M}\right)\right) + \mathbf{M}. \tag{8}$$

Overall, at the prior module of GLSSB, we capture global dependencies using a local-enhanced SSM. Because the SSM is better at learning global information, the subsequent module aims to handle more refined and complicated local dependencies.

**Local-Enhanced State Space Module**. Existing state space models [8, 12, 10] excels at capturing the causal processing of input data in long range dependencies. However, the unidirectional scan manner encounters difficulties in vision data to modeling non-causal relationships. To accommodate vision data, [65, 31, 41] process the input data from different 2D scan directions. However, these methods ignore the local invariants of vision data as shown in Fig. 2(c). In other word, the fixed scanning methods widen the distance between neighborhood data and snarl the causal relationships.

The most SSMs [31] can be regarded as the continuous linear time-invarian systems, we further introduce the a $e\left(\mathbf{L}_{\mathbf{p}}\right)$ augmented local bias, enhancing the original SSMs by preserving local 2D dependency. Following [58, 19], we propose a novel global-then-local state space:

$$\begin{aligned} h_t &= \bar{\mathbf{A}}h_{t-1} + \bar{\mathbf{B}}x_t, \\ y_t &= \mathbf{C}h_t + \mathbf{D}x_t + e\left(\mathbf{L}_{\mathbf{p}}\right), \end{aligned} \tag{9}$$

where $e\left(\mathbf{L}_{\mathbf{p}}\right)$ is independent of the hidden state space. Hence, the model can be computed in a simple way, given a feature $\mathbf{F}_g^{i-1} \in \mathbb{R}^{H \times W \times C}$ and illumination feature $\mathbf{L}_{\mathbf{p}} \in \mathbb{R}^{H \times W \times 5}$, we adopts the LayerNorm followed by our proposed LESSM to integrate the spatial long-term dependency. Following [10], the input feature are chunk into $\tilde{\mathbf{F}}_{\mathbf{1}}$ and $\tilde{\mathbf{F}}_{\mathbf{2}}$ in two branches. The first branch projects the feature into a linear layer, followed by a depth-wise convolution, SiLU activation function, accompanied by our proposed augmented local bias and LayerNorm. In the second branch, the features is also projected to a linear layer followed by the SiLU activation function. Finally, features

Table 1: Quantitative comparisons on LOL-V2-real, LOL-V2-syn, SMID, SDSD-indoor and SDSD-outdoor datasets. The best result is in red color while the second best result is in blue color.

| Methods | Ref. | LOL-V2-real | | LOL-V2-syn | | SMID | | SDSD-indoor | | SDSD-outdoor | | Complexity | |
|---|---|---|---|---|---|---|---|---|---|---|---|---|---|
| | | PSNR | SSIM | PSNR | SSIM | PSNR | SSIM | PSNR | SSIM | PSNR | SSIM | FLOPS | Param |
| RetinexNet | BMVC 2018 | 15.47 | 0.567 | 17.13 | 0.798 | 22.83 | 0.684 | 20.84 | 0.617 | 20.96 | 0.629 | 587.47 | 0.84 |
| DeepUPE | CVPR 2019 | 13.27 | 0.452 | 15.08 | 0.623 | 23.91 | 0.690 | 21.70 | 0.662 | 21.94 | 0.698 | 21.10 | 1.02 |
| SID | ICCV 2019 | 13.24 | 0.442 | 15.04 | 0.610 | 24.78 | 0.718 | 23.29 | 0.703 | 24.90 | 0.693 | 13.73 | 7.76 |
| KinD | MM 2019 | 14.74 | 0.641 | 13.29 | 0.578 | 22.18 | 0.634 | 21.95 | 0.672 | 21.97 | 0.654 | 34.99 | 8.02 |
| MIRNet | ECCV 2020 | 20.02 | 0.820 | 21.94 | 0.876 | 25.66 | 0.762 | 24.38 | 0.864 | 27.13 | 0.837 | 785.00 | 31.76 |
| EnGAN | TIP 2021 | 18.23 | 0.617 | 16.57 | 0.734 | 22.62 | 0.718 | 23.29 | 0.703 | 24.90 | 0.693 | 61.01 | 114.35 |
| Restormer | CVPR 2022 | 19.94 | 0.827 | 21.41 | 0.830 | 26.97 | 0.758 | 25.67 | 0.827 | 24.79 | 0.802 | 144.25 | 26.13 |
| SNR-Net | CVPR 2022 | 21.48 | 0.849 | 24.14 | 0.928 | 28.49 | 0.805 | 29.44 | 0.894 | 28.66 | 0.866 | 26.35 | 4.01 |
| QuadPrior | CVPR 2024 | 20.48 | 0.811 | 16.11 | 0.758 | 15.50 | 0.604 | 22.22 | 0.783 | 18.26 | 0.662 | / | / |
| MambaIR | ECCV 2024 | 21.25 | 0.831 | 25.55 | 0.929 | 27.07 | 0.774 | 28.97 | 0.884 | 29.75 | 0.861 | 60.66 | 4.30 |
| Retinexformer | ICCV 2023 | 22.80 | 0.840 | 25.67 | 0.930 | 29.15 | 0.815 | 29.77 | 0.896 | 29.84 | 0.877 | 15.57 | 1.61 |
| MambaLLIE | / | 22.95 | 0.847 | 25.87 | 0.940 | 29.26 | 0.818 | 30.12 | 0.900 | 30.00 | 0.869 | 20.85 | 2.28 |

from the two branches are aggregated with the element-wise product and then are projected back to input feature space by linear layer. The entire process can be delineated as:

$$\widetilde{\mathbf{F}}_1 = \mathrm{LN}\left(\mathrm{2DSSM}\left(\mathrm{SiLU}\left(\mathrm{DWConv}\left(\mathrm{Linear}\left(\mathbf{F}_1\right)\right)\right)\right) + \mathrm{Conv}(\mathbf{L}_p)\right),$$

$$\widetilde{\mathbf{F}}_2 = \mathrm{SiLU}\left(\mathrm{Linear}\left(\mathbf{F}_2\right)\right), \tag{10}$$

$$\widetilde{\mathbf{F}}_{\mathrm{out}} = \mathrm{Linear}\left(\widetilde{\mathbf{F}}_1 \odot \widetilde{\mathbf{F}}_2\right).$$

**Implicit Retinex-Aware Selective Kernel Module.** In our framework, we propose a Retinex-aware kernel selective mechanism (IRSK), where two coupled Retinex-aware priors are used to select the spatial context regions, the maximum and mean values of RGB images can be regarded as a rough illumination prior. IRSK constructs a sequence of depth-wise convolutions with an alterable kernel to select the feature with different receptive field, using a spatial selection mechanism by illumination prior. Inspired by LSKNet [29], for each of the feature maps from different selective kernel, a Sigmoid activation function is applied to obtain the individual illumination maps from illumination prior. Fig. 2(d) shows a detailed conceptual illustration of IRSK module where we intuitively demonstrate how the implicit Retinex-aware module works. The above process can be formulated as:

$$\widetilde{\mathbf{F}}_k = \widetilde{\mathbf{F}}_{\mathrm{out}}, \quad \widetilde{\mathbf{F}}_{k+1} = f_{\mathrm{DWconv}}^k\left(\widetilde{\mathbf{F}}_k\right). \tag{11}$$

The output of the Retinex-aware maps are concatenated with the input features via residual connections, followed by a depth-wise convolution, GELU activation function and convolution layer.

$$\{\mathbf{S}_1, \mathbf{S}_2\} = \mathrm{Chunk}\left(\mathrm{Sigmoid}\left(\mathrm{Conv}\left(\mathbf{L}_p\right)\right)\right), \tag{12}$$

$$\mathbf{F}_g = \mathrm{Conv}\left(\mathrm{GELU}\left(\mathrm{DWConv}\left(\sum_{k=1}^{K}\widetilde{\mathbf{F}}_k\mathbf{S}_k + \widetilde{\mathbf{F}}_{\mathrm{out}}\right)\right)\right). \tag{13}$$

## 4 Experiments

### 4.1 Benchmark Datasets and Implementation Details

**Datasets.** We employ five paired low light image datasets for evaluation, including LOL-V2-real [59], LOL-v2-syn [59], SMID [4], SDSD-indoor [48] and SDSD-outdoor [48] datasets. Therein, LOL-V2-real contains 689 low-normal light paired images for training and 100 pairs for testing; LOL-V2-syn includes 900 paired images for training and the 100 pairs for testing; Besides, SMID is composed of the 15763 short-long exposure paired images for training and the remaining images for testing; SDSD-indoor and SDSD-outdoor are all subsets of SDSD dataset (the static version), which extract the paired images from 62 and 116 pairs for training, and the left 6 and 10 pairs for testing.

**Implementation Details**. We implement MambaLLIE in PyTorch [40] on a server with the 4090GPUs. Random cropping the image pairs into $128 \times 128$ patches as training samples, data augmentation is performed on the training samples such as rotation and flipping. The batch size is 8. In terms of optimization procedure, Adam [24] is adopted as the optimizer with $\beta_1 = 0.9$ and $\beta_2 = 0.999$; The training iterations is set to $1.5 \times 10^5$. The initial e learning rate is set to $2 \times 10^{-4}$ and steadily decreased by by the cosine annealing scheme. The loss criterion is mean absolute error (MAE), thus peak signal-to-noise ratio (PSNR) and structural similarity (SSIM) [52] is selected as the evaluation metrics for the paired datasets.

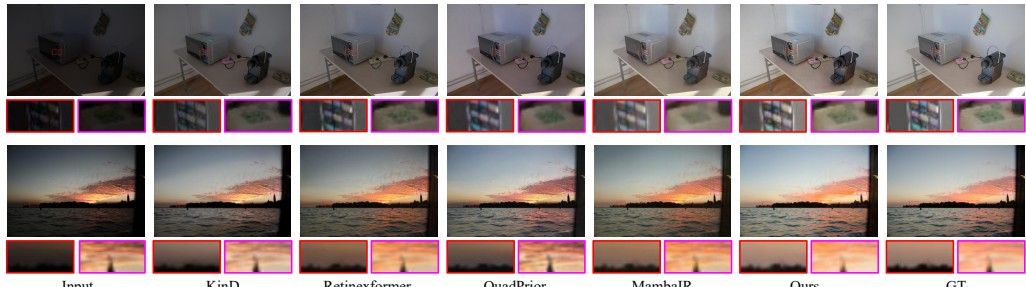

| Input | KinD | Retinexformer | QuadPrior | MambaIR | Ours | GT |

Figure 3: Qualitative comparison with previous methods on LOL-V2-real and LOL-V2-syn datasets. Our MambaLLIE effectively enhances the illumination and preserves the color.

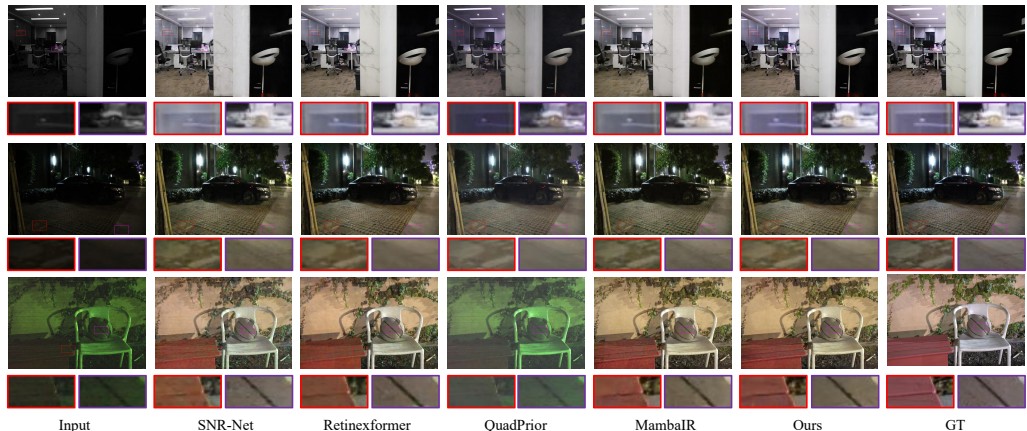

| Input | SNR-Net | Retinexformer | QuadPrior | MambaIR | Ours | GT |

Figure 4: Qualitative comparison with previous methods on SMID, SDSD-indoor and SDSD-outdoor datasets. Our MambaLLIE restore the texture and color under challenging degradation, such as the wooden bench and reflective glass.

## 4.2 Main Results on Benchmarks.

**Quantitative Comparison.** As shown in Tab. 1, we evaluated the performance of our MambaLLIE against 11 SOTA image enhancement methods, including RetinexNet [53], DeepUPE [49], SID [4], KinD [64], MIRNet [63], EnGan [21], Restormer [62], SNR-Net [56], QuadPrior [51], MambaIR [16] and Retinexformer [3]. Our MambaLLIE demonstrates superior performance than SOTA methods on the adopted benchmark datasets in terms of PSNR and SSIM, while achieves comparable results of SSIM with the SOTA methods in LOL-V2-real and SDSD-outdoor. Therein, when the parameters are roughly similar, our MambaLLIE achieves an average improvement of 0.2dB on benchmark datasets compared to the Transformer based SOTA method, *i.e.* Retinexformer. Compared with the earlier Transformer based SNR-Net, MambaLLIE outperforms it by average 1dB PSNR on the all datasets. When compared to the MambaIR, MambaLLIE achieves 1.7, 0.32, 2.19, 1.15 and 0.25dB PSNR improvements on the adopted datasets, respectively. Besides, Our MambLLIE gains the improvements over 7 dB on all datasets than traditional Retinex-based models, such as RetinexNet, DeepUPE and KinD.

**Qualitative Comparison.** Figs. 3 & 4 report the vision results for comparing our method with latest the SOTA methods and traditional Retinex-based models. Existing methods suffer from insufficient illumination and fail to restore the details as shown in Fig. 3. As we can see, color distortion and image degradation also affect the enhanced results of previous methods in Fig. 4, yet our MambaLLIE not only enhances brightness but also faithfully preserves colors with reference to ground truth images, all while restoring the details.

## 4.3 Real World Experimental Evaluation

Enhancing low-light images in real-world scenarios is exceptionally challenging because, in addition to benefiting downstream tasks such as dark object detection, the enhanced images must also be visually pleasing to human perception.

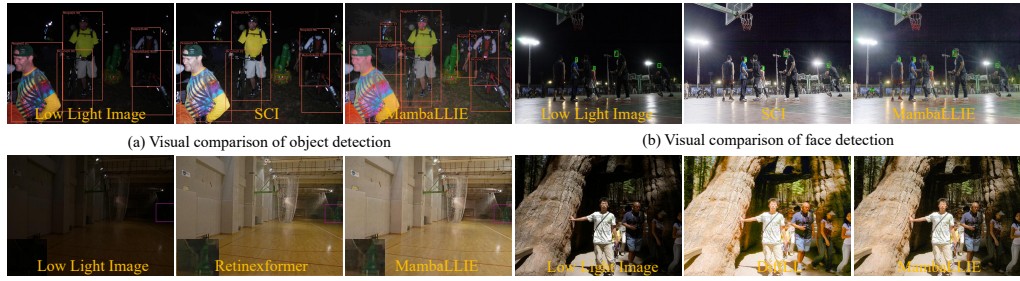

(a) Visual comparison of object detection      (b) Visual comparison of face detection

(c) Toy example of user study      (d) Visual comparison of unpaired dataset

Figure 5: Visual comparison of our MambaLLIE with recent SOTA methods. (a) Qualitative comparison on object detection, (b) Qualitative comparison on face detection, (c) Toy example of user study, (d) Qualitative comparison of unpaired dataset.

Table 2: Low light object detection results on the ExDark dataset. The best result is in red color while the second best result is in blue color.

| Methods | Bicycle | Boat | Bottle | Bus | Car | Cat | Chair | Cup | Dog | Motor | People | Table | Mean |
|---|---|---|---|---|---|---|---|---|---|---|---|---|---|
| RetinexNet | 0.790 | 0.741 | 0.743 | 0.908 | 0.820 | 0.665 | 0.651 | 0.750 | 0.721 | 0.703 | 0.784 | 0.556 | 0.736 |
| EnGAN | 0.733 | 0.710 | 0.687 | 0.892 | 0.786 | 0.675 | 0.656 | 0.650 | 0.741 | 0.657 | 0.731 | 0.528 | 0.704 |
| KinD | 0.800 | 0.721 | 0.788 | 0.919 | 0.822 | 0.718 | 0.672 | 0.771 | 0.775 | 0.736 | 0.803 | 0.555 | 0.757 |
| ZeroDCE | 0.806 | 0.750 | 0.762 | 0.914 | 0.837 | 0.681 | 0.677 | 0.769 | 0.788 | 0.728 | 0.801 | 0.535 | 0.754 |
| SCI | 0.821 | 0.742 | 0.749 | 0.916 | 0.846 | 0.695 | 0.690 | 0.784 | 0.756 | 0.758 | 0.810 | 0.555 | 0.760 |
| SNR-Net | 0.802 | 0.721 | 0.750 | 0.932 | 0.840 | 0.694 | 0.677 | 0.758 | 0.763 | 0.755 | 0.789 | 0.559 | 0.753 |
| Retinexformer | 0.809 | 0.769 | 0.753 | 0.914 | 0.814 | 0.688 | 0.689 | 0.763 | 0.766 | 0.769 | 0.805 | 0.543 | 0.757 |
| MambaIR | 0.803 | 0.763 | 0.752 | 0.903 | 0.830 | 0.687 | 0.684 | 0.761 | 0.721 | 0.738 | 0.813 | 0.556 | 0.751 |
| MambaLLIE | 0.802 | 0.764 | 0.779 | 0.926 | 0.846 | 0.701 | 0.692 | 0.800 | 0.781 | 0.751 | 0.812 | 0.560 | 0.768 |

Table 3: Face detection results on the Dark face dataset. The best result is in red color while the second best result is in blue color.

| Methods | Low light image | SCI | Retinexformer | MambaIR | Ours |
|---|---|---|---|---|---|
| mAP | 0.461 | 0.483 | 0.486 | 0.482 | 0.491 |

**Low Light Object Detection.** We utilized ExDark dataset [32] to compare the enhancement of preprocessing methods for high-level vision tasks. There are 7363 challenging low light images annotated with 12 bounding box classes, of which 5,890 for training and 1,473 for testing. Note that all supervised methods were pretrained on the LOL-V2-syn dataset, the low light image underwent different enhancement methods and then finetuned YOLOv3 [43] as the object detector. As shown in Tab. 2, our methods achieved the best average result compared with other models, and yielded the best results on Car, Chair, Cup, People and Table classes. Fig. 5(a) further reported the visual comparison, compared with suboptimal preprocessing method SCI, detector through our MambaLLIE can detect the objects in extreme dark regions including two persons and a chair, while other methods failed.

**Face Detection** We investigate the performance of low-light image enhancement methods on face detection in the dark. We use the DARK FACE dataset [60] and randomly sample 300 images for evaluation. The RetinaFace [44] is used as the face detector and fed with the results of different LLIE methods. We show the results of different methods in Fig. 5(b) and Tab. 3. In general, MambaLLIE achieves the better mAP score and visual detection result. Please note that the effectiveness of face detection in low light conditions depends not only on the quality of the enhancement results but also on the specific face detection algorithm employed. We utilize the pre-trained RetinaFace model to assess the performance of different low light image enhancement methods to some extent.

**User Study.** We conducted a user study to evaluate the human visual perception quality of the enhanced results in challenging scenarios. Due to the lack of the ideal reference for training, we selected the pretrained model from the benchmarks to enhance the photos. There are 7 random selected low light images from the benchmarks and ExDark datasets under different lighting conditions. Human perception primarily focuses on the presence of global visual effect, local detail, color distortion (noise), which significantly reflect the quality of the enhanced images. Thus, We assigned ratings on a scale of 1 (worst) to 5 (best), evaluating the quality of the enhancements in terms of overall rating, local detail and color distortion(noise), respectively. Overall, 70 participants were invited to assess the visual quality. The average scores are reported in Tab. 4, our MambaLLIE

Table 4: User study on the challenging low light image enhancement.

| Methods | RetinexNet | EnGAN | SCI | QuadPrior | SNR-Net | Retinexformer | MambaIR | MambaLLIE |
|---|---|---|---|---|---|---|---|---|
| Overall Rating | 3.093 | 3.314 | 3.943 | 3.014 | 3.821 | 4.100 | 3.857 | **4.243** |
| Local Detail | 2.871 | 3.143 | 3.686 | 3.129 | 3.779 | 3.950 | 3.629 | **4.129** |
| Artifacts and noise | 2.914 | 3.164 | 3.776 | 2.929 | 3.657 | 3.971 | 3.750 | **4.100** |

Table 5: Perceptual evaluation results on the unpaired datasets

| Methods | LIME | | VV | | NPE | | MEF | | DICM | |
|---|---|---|---|---|---|---|---|---|---|---|
| | MUSIQ | NIMA | MUSIQ | NIMA | MUSIQ | NIMA | MUSIQ | NIMA | MUSIQ | NIMA |
| Retinexformer | **58.66** | 4.85 | 58.96 | 4.63 | **56.98** | 4.81 | 54.27 | 4.89 | 55.69 | 4.84 |
| MambaIR | 56.31 | 4.70 | 59.29 | 4.70 | 56.38 | 4.75 | 53.84 | 4.88 | 57.01 | 4.84 |
| DiffLL | 55.39 | 4.66 | 58.62 | 4.55 | 53.54 | 4.65 | 52.14 | 4.91 | 55.77 | 4.82 |
| Ours | 58.42 | **4.86** | **60.22** | **4.78** | 56.70 | **4.82** | **55.01** | **4.93** | **57.16** | **4.92** |

achieves the best score in the involved voting aspects. Fig. 5(c) shows the toy example of user study, which display the input and the random enhanced results and local details by different algorithms.

**Perceptual Evaluation.** We compare two non-reference perceptual metrics, MUSIQ [23] and NIMA [6], on five unpaired datasets, including LIME [17], VV [47], NPE [50], MEF [37], and DICM [26]. Experimental evaluations in Tab. 5 show the superiority of our method over SOTAs with better perceptual evaluation in most comparisons, in terms of the NIMA scores, our method also achieves competitive results across all datasets. Experimental comparisons against the diffusion-based model DiffLL [20] also underline our method's robustness. These results are visually supported in Fig. 5(d), where our method demonstrates qualitative improvements that align well with its perceptual metrics. Overall, the evaluations illustrate the superiority of our enhancement approach, achieving enhanced perceptual quality and setting a new LLIE benchmark on unpaired datasets.

## 4.4 Ablation Study

**Implicit Retinex-Aware Framework.** We compare the improvement of using a implicit Retinex-aware model with the end-to-end and explicit Retinex-aware models. Specifically, Baseline-1 is a simple variant of our MambnaLLIE by removing Retinex-aware guidance, directly uses the standard vision state space module (VSSM) to process flattened vision data in our proposed UNet-shaped framework, following the *Norm → VSSM → Norm → channel attention layer* flow as referenced in [16]. Baseline-2 is designed to estimate the illumination map and then light up the low light image by element-wise multiplication, namely aims to estimate the illumination map instead of directly predicting the enhanced image, and then restores the enhanced result by $N = \tilde{R} \cdot \tilde{I}$. Tab. 6 reveals our implicit Retinex-Aware framework significantly outperforms Baseline-1 with the improvement of 1.25dB in PSNR, while achieving a PSNR enhancement of 1.00 dB compared to Baseline-2 on SDSD-indoor dataset.

**Global-then-Local State Space.** As the core component, our GLSSB comprises the LESSM and IRSK. We demonstrate the effect of each component through ablation study. For example, The results on SDSD-indoor dataset, presented at Tab. 6, indicate that our LESSM achieves improvements of 0.33 dB and 0.08 dB in PSNR compared to Baseline-1 and Baseline-2, respectively, which utilize vanilla state space blocks. Additionally, our IRSK produces PSNR enhancements of 0.96 dB, 0.74 dB, and 0.63 dB compared to Baselines and when only applying LESSM. Our full version indicating that although LESSM improves the vanilla SSM with local enhanced modeling ability, IRSK should be considered for further improvements, when GLSSB integrats LESSM and IRSK, our MambaLLIE achieves the highest PSNR and SSIM.

**Selective Kernel Behavior.** We further investigate the kernel selection behavior in our MambaLLIE as shown in Fig. 6. We find the implicit Retinex-aware selection pattern tend to learn two independent positive and negative illumination, resulting in complementary features. Compared with explicit Retinex-based methods, our IRSK can guide from a flexible deeper neural representation. The quantitative results are reported in Tab. 7. Different with LSKNet [29], we put small kernels in front and larger kernels in higher levels. This is because object detection needs larger receptive field, thus adopts a sequence of depth-wise convolutions with growing kernel and increasing dilation, while has to introduce a lots of padding. But image enhancement may suffers from padding operation at the edge of the image, especially upsampling further expands the padding values. Thus, the the former

Table 6: Effects of design choices.

| Methods | Params(M) | FLOPS(G) | LOL-V2-real | | SDSD-indoor | | SDSD-outdoor | |
|---|---|---|---|---|---|---|---|---|
| | | | PSNR | SSIM | PSNR | SSIM | PSNR | SSIM |
| Baseline-1 | 2.14 | 18.39 | 22.06 | 0.834 | 28.87 | 0.865 | 28.86 | 0.852 |
| Baseline-2 | 2.14 | 18.39 | 21.28 | 0.812 | 29.12 | 0.862 | 28.96 | 0.841 |
| Ours w/o LESSM | 2.26 | 20.64 | 21.83 | 0.846 | 29.83 | 0.889 | 29.20 | 0.866 |
| Ours w/o IRSK | 2.19 | 19.94 | 22.37 | 0..845 | 29.20 | 0.887 | 28.97 | 0.857 |
| Ours | 2.28 | 20.85 | **22.95** | **0.847** | **30.12** | **0.900** | **30.00** | **0.869** |

Table 7: Effects of IRSK.

| Kernel Sizes | Params | FLOPS | PSNR | SSIM |
|---|---|---|---|---|
| 3*3 | 2.25 | 20.47 | 29.55 | 0.899 |
| 5*5 | 2.31 | 21.23 | 29.48 | 0.896 |
| 5*7 | 2.35 | 21.79 | 28.88 | 0.892 |
| 5*3 | 2.28 | 20.85 | 29.31 | 0.892 |
| 3*5 (Ours) | 2.28 | 20.85 | **30.12** | **0.900** |

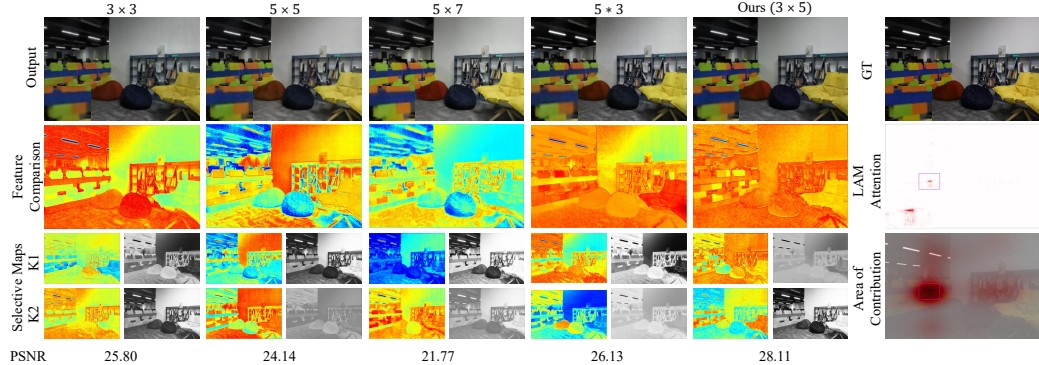

Figure 6: The details of selective kernel behaviour, the LAM visualization [14] demonstrates influence of similar local information is higher than that of global dependence, our local-enhanced strategy underscores the feature. Besides, the larger receptive fields can provide globally consistent results.

small kernels can quickly focus on local information and the the latter kernels contain larger receptive fields for better feature fusion.

# 5    Limitation and Discussion

We adopt an implicit Retinex-Aware guidance within a global-then-local state space framework to address global insufficient illumination and local degradation for low light enhancement. However, our approach has several limitations. 1) Unlike end-to-end methods, our technique requires the design of a reasonable illumination prior, which relies on prior experience. 2) Most existing enhancement models, including ours, primarily focus on mean square error and use PSNR and SSIM to evaluate image quality. To mitigate inherent biases in these metrics, we conducted additional real-world experimental evaluations to reconcile the bias and further validate the effectiveness of our approach.

# 6    Conclusion

In this paper, we introduced a novel state space-based model, MambaLLIE. Our proposed core of GLSSB effectively combines global and local information by implicit Retinex-aware selective kernel into global-then-local state space. Extensive experiments on benchmarks, low light object detection, face detection, user study and perceptual evaluation demonstrate that our framework consistently achieves the best performance. Our future work is to address the dual challenges of local redundancy and global dependencies in low light video enhancement via efficient state space modeling. Broader impact and more visual results, please refer to Appendix A and Appendix B.

## Acknowledgements

This work was partially supported by the National Science Fund of China, Grant Nos. 62072242 and 62361166670. We sincerely appreciate the valuable feedback provided by the anonymous reviewers.

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

## A  Broader Impact.

Low light image enhancement is the classical task that improves the quality of degraded images, exhibiting the promising value of research and application. Our proposed global-then-local state space enhances the feature extraction ability by integrating implicit Retinex-aware strategy. We believe our method has the potential to advance other low-level tasks and may inspire future research in state space models. However, there could be negative effects brought by the proposed method. For example, the inevitable deviations of training data distribution, the generated results for the real world scenarios may exist the color deviation.

## B  More Results.

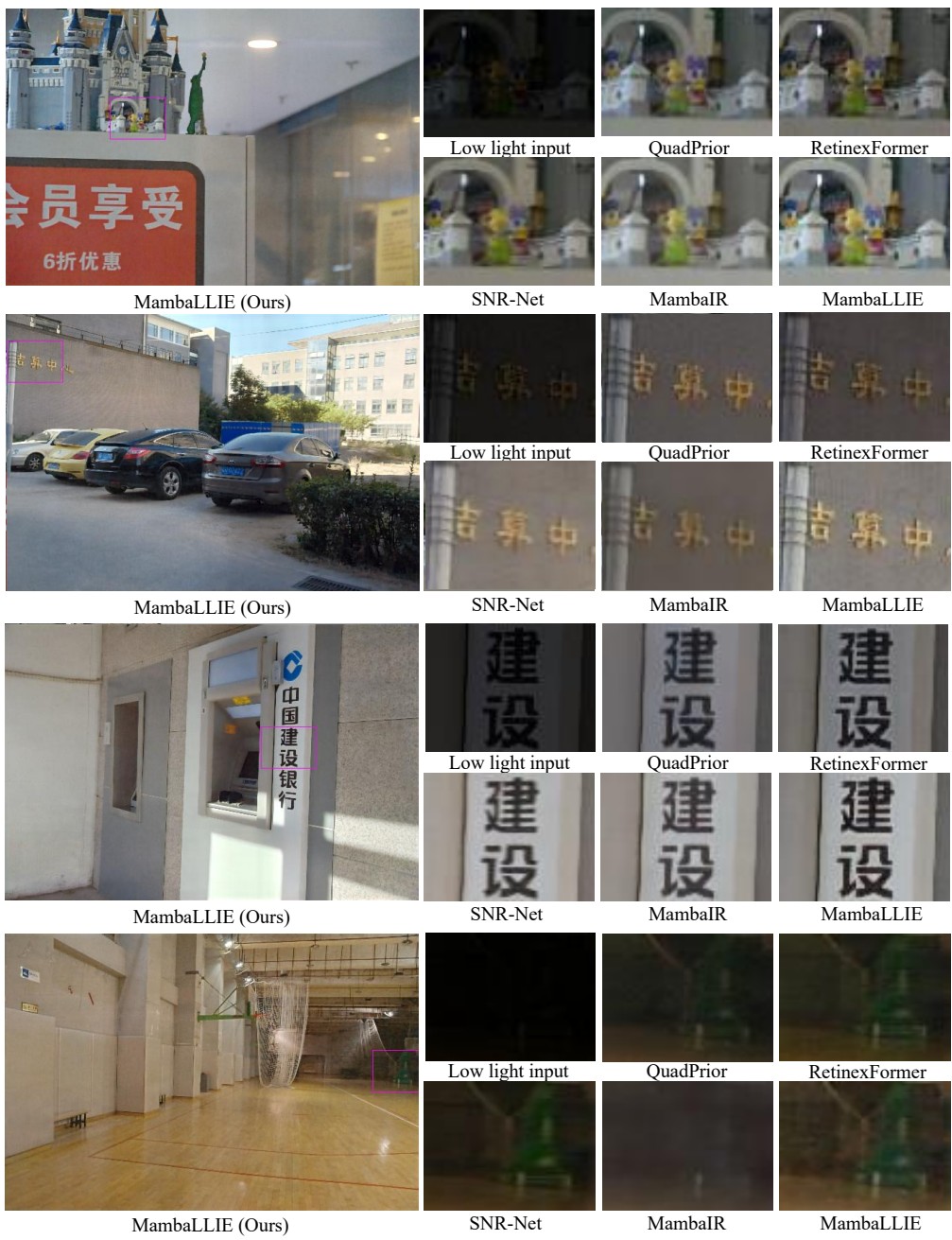

Figure 7: More qualitative comparisons with SOTAs. (Zoom in for best view)

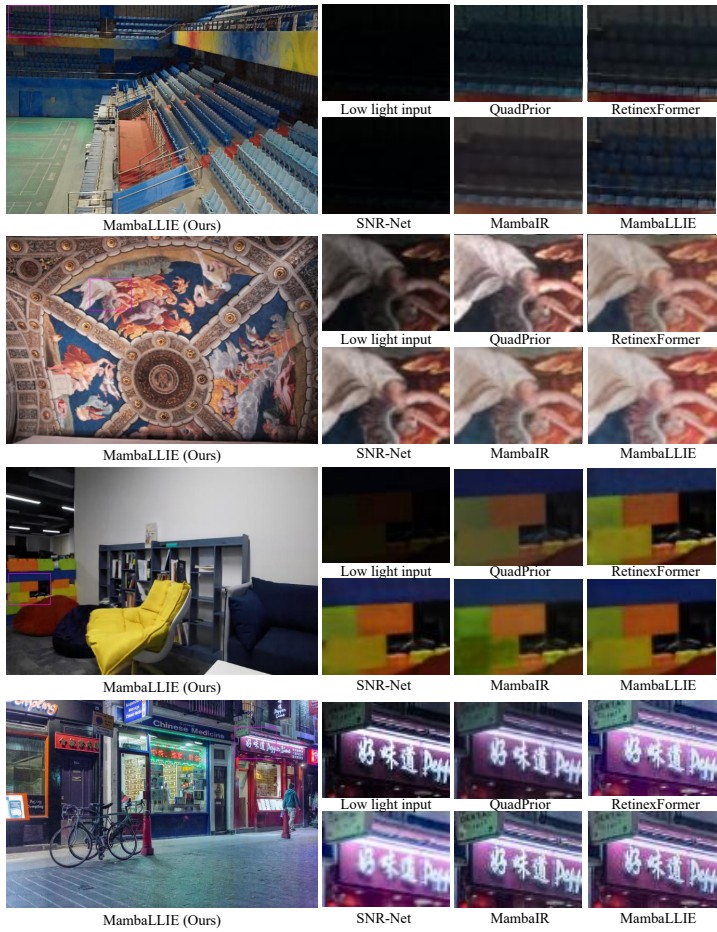

Figure 8: More qualitative comparisons with SOTAs. (Zoom in for best view)

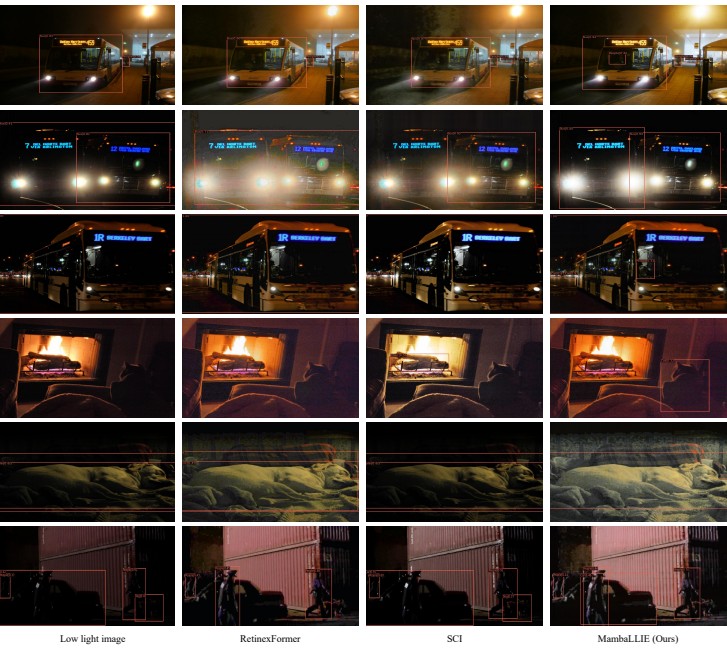

Figure 9: Object detection qualitative comparisons with SOTAs. (Zoom in for best view)

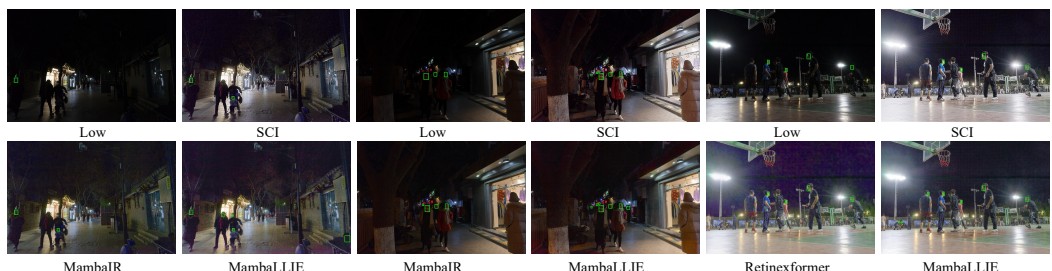

Figure 10: Qualitative comparisons on face detection performance. (Zoom in for best view)

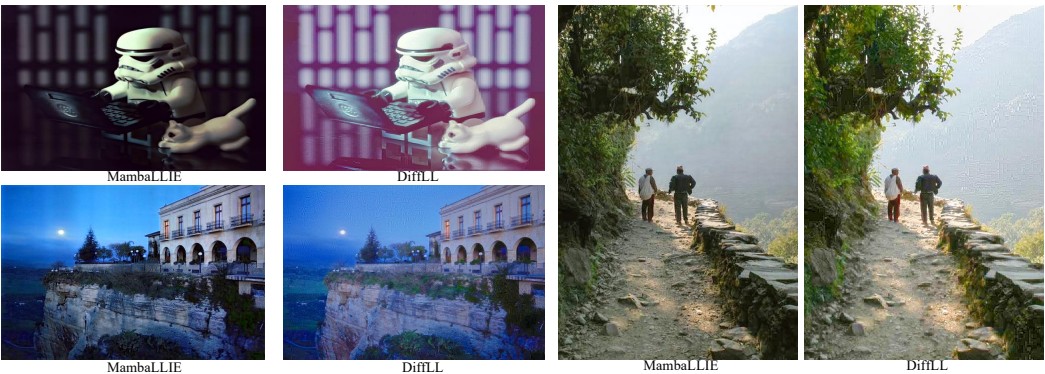

Figure 11: Qualitative comparisons of MambaLLIE and DiffLL. (Zoom in for best view)

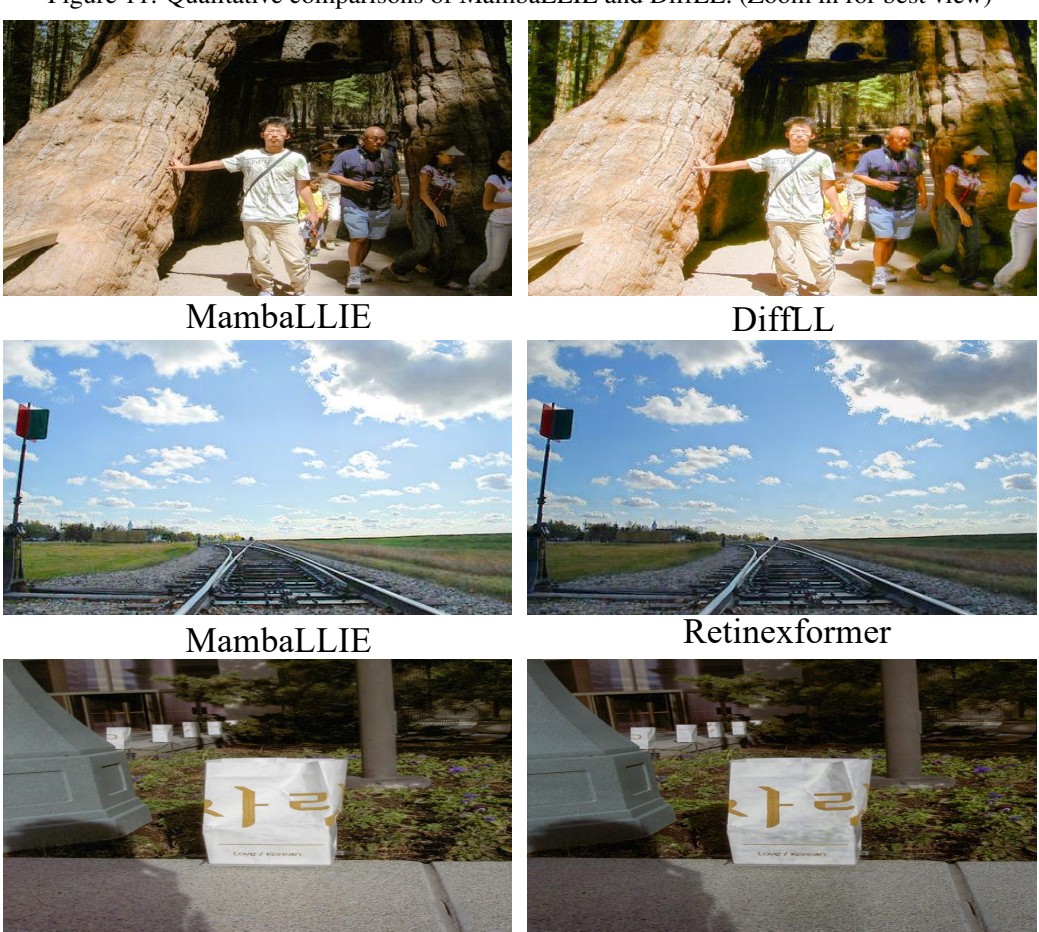

Figure 12: Qualitative comparisons on unpaired datasets. (Zoom in for best view)

