# OpenReview forum: "MambaLLIE: Implicit Retinex-Aware Low Light Enhancement with Global-then-Local State Space"
_NeurIPS.cc/2024/Conference — NeurIPS 2024 poster_

### Official Review · Reviewer_1rUY · 2024-06-23

**Soundness:** 3
**Presentation:** 3
**Contribution:** 3
**Rating:** 8
**Confidence:** 5

**Summary:**

This paper proposes a Mamba-based framework, namely MambaLLIE, for low-light image enhancement. Specifically, the authors claim that they have two technical contributions:

(i) A global-then-local state space block that integrates a local-enhanced state space module and an implicit Retinex-aware selective kernel module to capture intricate global and local dependencies.

(ii) An implicit Retinex-aware selective kernel mechanism to guide deeper neural representations and segregate them into independent positive and negative illumination components before integrating them.

**Strengths:**

(i) The idea is novel and interesting. Mamba is a scorching research topic in computer vision now and has shown promising performance in many high-level vision tasks like image recognition, object detection, image segmentation, etc. But until now, there have been fewer efforts dedicated to the low-light image enhancement task, a small low-level topic. Thus, it is very exciting to see this wonderful work.

(ii) The presentation is well-dressed. I like the style of the figures in this paper. For example, the teaser figure can clearly show the advantages in larger receptive fields of the proposed MambaLLIE than previous Transformer-based methods and Mamba-based image restoration methods. For another example, the pipeline in Figure 2 can also clearly show the workflow and the details of each submodule of the whole framework.

(iii)  The performance is good and solid. The proposed MambaLLIE significantly outperforms the state-of-the-art methods on six benchmarks including LOL-v2-syn, LOL-v2-real, SMID, SDSD-indoor, and SDSD-outdoor, as reported in Table 1. The visual comparisons in Figure 3 also show the effectiveness of the proposed method. Looks good.

**Weaknesses:**

(i) The motivation is not clear. To be specific, why use Mamba for low-light image enhancement? The authors claim that the Mamba can help capture long-range dependences. However, Transformer architecture can also model the long-range dependences. Why use Mamba instead of Transformer? Also, why design the space state module like this? The insight of the proposed Mamba should be analyzed in more detail.

(ii) A technique should be explained, i.e., what is the scanning operation in the state space module? The authors mention the scan in Mamba but I could not find any mathematical formula or text description in the paper to explain the computation process of this scan.

(iii) In table 1, although the performance of the proposed MambaLLIE is better than the state-of-the-art Transformer-based method Retinexformer, its computational complexity and memory cost are larger than those of Retinexformer. For example, the FLOPS and the Params are 33% and 38% higher. In addition, what about the comparison of performance on the MIT-Adobe-FiveK dataset (sRGB output mode)?

(iv) In Figure 4, it seems that Retinexformer achieves more visually pleasant results. For example, in the third line of Figure 6, Retinexformer reconstructs the complete edge of the redwood in the left zoomed-in patch but the proposed MambaLLIE fails. Thus, it is better to compare MambaLLIE and the state-of-the-art method on unpaired datasets such as the LIME, NPE, MEF, DICM, and VV datasets.

(v) The source code and pre-trained models have not been submitted. The reproducibility cannot be checked.

**Questions:**

I am curious about the training time comparison between the proposed MambaLLIE and other methods.

**Limitations:**

Yes, the authors have analyzed the limitations in the main paper.

---

> ### Author Rebuttal · Authors · 2024-08-07
>
> **Q1: Why use Mamba for low-light image enhancement?**
>
> **A1:** Our **Motivation** aims to take into account both global and local image restore. As we know, the low-light enhancement task faces challenges of global color degeneration and local noise disturbance. Global color degeneration suffer from the decrease of illumination, which tends to introduce a global-aware estimator, while the local degradation may contains the large or small regions. Hence, we intuitively introduce Mamba [8] into low light enhancement task, which excels at capturing the global dependencies of input data to restore. Compared with previous CNN and Transformer-based methods, Mamba-based model has exhibit promising performance with **linear or near-linear scaling complexity** in image super-resolution, image segmentation, etc., but still has the **suboptimal results** in low light enhancement task.
>
> **Q2: Why use Mamba instead of Transformer?**
>
> **A2:** We chose Mamba over Transformers due to its efficient long-range dependency modeling with linear complexity. Transformers can model long-range dependencies, but they typically involve higher computational complexity and memory costs.
>
> **Q3: Why design the space state module like this?**
>
> **A3**: Most prior VSSMs use the different directional scan in their sequential state, which is necessary for strong performance on visual tasks. However, it has been empirically observed that CNN seem to work fine for 2D dependency of vision data. Ours local-enhanced design essentially introduces the local invariance into state space model, which can integrate the existing directional scan with our local-enhanced term into VSSM. By additional convolving 2D information into directional scan, our method ensures a closer arrangement of relevant local tokens, enhancing the capture of local dependencies. This technique is depicted in Figure 2 (c). Next, we follow the selection mechanism by parameterizing the VSSM parameters based on the input, allows the model to filter out irrelevant information and remember relevant information indefinitely, as pointed in [8]. Besides, our local-enhanced term can be regarded as a local consistency constraint for the state space for vision data, guiding the module to learn neighborhood features and thus enhancing robustness, literatures [47], [C1] likewise added constraints of local term to a state–space model as equation 10 and developed the estimate performance.
>
> **Q4: A technique should be explained, i.e., what is the scanning operation in the state space module?**
>
> **A4:** To allow Mamba to process 2D images, the feature map needs to be flattened before being iterated by the state space. Therefore, the unfolding strategy is particularly important. In this work, we follow [38], which uses scans in four different directions to generate scanned sequences.
>
> **Q5: In table 1, although the performance of the proposed MambaLLIE is better than the state-of-the-art Transformer-based method Retinexformer, its computational complexity and memory cost are larger than those of Retinexformer. For example, the FLOPS and the Params are 33\% and 38\% higher.**
>
> **A5:** Reducing the computational complexity and memory cost is not the key part of MambaLLIE. Our MambaLLIE targets addressing the contradiction between global and local context interaction in the VSSM and proposes the flexible and effective IRSK for the LLIE task to achieve a larger receptive field in terms of global and local information. Besides, we significantly reduced the computational complexity and memory cost compared with our competitor MambaIR and kept comparable costs with other competitors. Additionally, we acknowledge that RetinexFormer is an outstanding work, maintaining a good balance in terms of parameter and perfermance. Our method slightly surpasses it in terms of performance.
>
> **Q6: What about the comparison of performance on the MIT-Adobe-FiveK dataset (sRGB output mode)?**
>
> **A6:** Following your valuable suggestion, we conducted the experiments of MambaLLIE and recent SOTA methods on the MIT-Adobe-FiveK dataset in Table R5, which verifies the comparable results of our method with RetinexFormer and surpasses the performance of other SOTA models. Figure R5 further shows the qualitative comparisons of MambaLLIE and RetinexFormer, where ours can remarkably improve the underexposure that appeared in RetinexFormer's result.
>
> **Q7: In Figure 4, it seems that Retinexformer achieves more visually pleasant results. For example, in the third line of Figure 6, Retinexformer reconstructs the complete edge of the redwood in the left zoomed-in patch but the proposed MambaLLIE fails. Thus, it is better to compare MambaLLIE and the state-of-the-art method on unpaired datasets such as the LIME, NPE, MEF, DICM, and VV datasets.**
>
> **A7:** Please refer to Table R4. We compare two non-reference perceptual metrics, MUSIQ and NIMA, on five unpaired datasets, including LIME, VV, NPE, MEF, and DICM. Experimental evaluations show the superiority of our method over SOTAs with better perceptual evaluation in most comparisons.
>
> **Q8: The source code and pre-trained models have not been submitted.**
>
> **A8:** Now, our code and the pretrained model are released on our project page. Please refer to our project page.
>
> **Q9: Training time comparison**
>
> **A9:** For example, training on the LOLV2 real dataset takes approximately 19 hours using PyTorch on a server equipped with 4090 GPUs, while Retinex tasks approximately 22 hours and MambaIR tasks approximately 27 hours . For the corresponding training parameters, please refer to lines 205-208 of the manuscript.

---

> > ### Comment · Reviewer_1rUY · 2024-08-07
> > **Response to the author rebuttal**
> >
> > Thanks for your response. Most of my concerns have been addressed except one.
> >
> > You claim that `Mamba over Transformers due to its efficient long-range dependency modeling with linear complexity. Transformers can model long-range dependencies, but they typically involve higher computational complexity and memory costs.`
> >
> > However, this is not true. Many Transformers also enjoy linear computational complexity, e.g., Retinexformer. It takes much shorter time to train than the Mamba-based methods.
> >
> > Anyway, this is a good attempt to explore the potential of Mamba in LLE. Thus, I decided to raise my score to `strong accept` to support you. Thanks for your effort.

---

> > > ### Author Response · Authors · 2024-08-08
> > >
> > > We sincerely appreciate your recognition of our work and effort.  YES, You are right,  as discussed in [3], *"O(IG-MSA) of Retinexformer is linear to the spatial size, mainly comes from the k computations of the two matrix multiplication of attention."* Compared to the complexity of the global MSA used by previous Transformer methods, Retinexformer significantly reduces the computational complexity of attention while enhancing performance in LLIE tasks. Besides, our MambaLLIE inherits the linear complexity of SSM for visual representation learning, and a series of improvements in state space and Retinex-aware module are adopted to improve the performance, which have not significant and even negligible impact on overall complexity.
> > >
> > > Finally, we once again extend our gratitude to the reviewer for the constructive feedback during the review and rebuttal stages, giving a final score of **strong accept** for our manuscript. The insightful comments on complexity remind us to maintain rigorous discussions and analyses in our future work.

---

### Official Review · Reviewer_1bAi · 2024-07-01

**Soundness:** 2
**Presentation:** 3
**Contribution:** 2
**Rating:** 4
**Confidence:** 5

**Summary:**

This paper presents MambaLLIE, an implicit Retinex-awre low light image enhancement framework with modified state space blocks. Specifically, a global-then-local state space block (GLSSB) is designed, which incorporates a local-enhanced state space module (LESSM) and an implicit Retinex-aware selective kernel (IRSK) module. By enhancing the original SSMs with local bias, the proposed MambaLLIE outperforms state-of-the-art CNN and Transformer-based methods.

**Strengths:**

1.	The main idea of this paper is well illustrated.
2.	Experimental results and user studies demonstrate that the paper achieves better performance than the compared approaches.

**Weaknesses:**

1. The novelty of this paper is limited. Like many Transformer-based works that embed local modeling capabilities in ViTs, the main contribution of this paper is incorporating local dependencies into SSMs. However, the advantages of the proposed GLSSB compared to ViTs and original SSMs have not been thoroughly analyzed. Furthermore, given the authors' design, the proposed GLSSB should have advantages in many other visual tasks, which need to be demonstrated through more experiments.

2. The writing of this paper needs improvement as there are many unclear descriptions. For example, how is the illumination prior in the paper obtained, how does the proposed IRSK module work specifically, and what are the specific settings for the different variants in the ablation study?

3. Some state-of-the-art works have not been compared (e.g., DiffLL[1]), and the experimental results in this paper do not seem to outperform them.

4. As the low-light enhancement task typically encounters issues like color distortion or other visual artifacts, the authors should compare more perceptual evaluation metrics. Merely reporting PSNR and SSIM is not sufficient.

[1] Jiang et al, Low-Light Image Enhancement with Wavelet-based Diffusion Models. Siggraph asia 2023.

**Questions:**

Please refer to the `Weaknesses'.

Moreover, it appears that the color in the last column of Table 2 is incorrectly marked.

**Limitations:**

The authors have discussed the limitations in the paper.

---

> ### Author Rebuttal · Authors · 2024-08-07
>
> **Q1: About novelty and the advantages of GLSSB.**
>
> **A1:** We argue that the GLSSB is novel in terms of _*the new exploration of the vision state space model*_ and _*technical improvements for the Retinex-aware low light enhancement task*_.
>
> Our **Motivation** aims to take into account both global and local image restore. As we know, the low-light enhancement task faces challenges of global color degeneration and local noise disturbance. Global color degeneration suffer from the decrease of illumination, which tends to introduce a global-aware estimator, while the local degradation may contains the large or small regions. Hence, we intuitively introduce Mamba [8] into low light enhancement task, which excels at capturing the global dependencies of input data to restore. Compared with previous CNN and Transformer-based methods, Mamba-based model has exhibit promising performance with **linear or near-linear scaling complexity** in image super-resolution, image segmentation, etc., but still has the **suboptimal results** in low light enhancement task. We posit that the existing state space model is limited in modeling local dependencies, leading to a failure in restoring details. Besides, end-to-end models may exhibit suboptimal performance in low-light enhancement tasks because they do not consider illumination priors.
>
> Based on the two aforementioned assumptions, we propose the local-enhanced state space module (LESSM) and implicit retinex-aware selective kernel module (IRSK), which is the modeling of local and global degradation coupled in our framework for low light enhancement task. Therefore, we believe our novelty and contribution is on the par with the SOTA methods.
>
> Therefore, we believe our novelty and contribution are on par with the SOTA methods.
>
> Since IRSK is specially designed for low light enhancement task, We replaced the VSSM of MambaIR with LESSM to demonstrate the effectiveness of ours.  As shown in Figure R1 and Table R1, we demonstrate the effectiveness of the proposed LESSM in MambaIR on image restoration tasks, indicating that our approach also has a positive impact on other low-level vision tasks, as discussed by \textbf{Reviewer 1bAi}. While the differences between LESSM and the vanilla VSSM may appear relatively small in description, each has its key motivations and shows significant improvements on common benchmarks over the previous models. We believe that simplicity combined with effectiveness is one of the most favored traits in the field of computer vision.
>
> **Q2: How is the illumination prior in the paper obtained and how does the proposed IRSK module work specifically.**
>
> **A2:** Sorry for the confusion caused. In our framework, we propose a Retinex-aware kernel selective mechanism, where two coupled Retinex-aware priors are used to select the spatial context regions, the maximum and mean values of RGB images can be regarded as a rough illumination prior. We use it in a implicit manner instead of restore the illumination map or use a single illumination feature for feature guidance. Base on this,  we clarify the work process of IRSK as stated in Lines 184-194.
>
>
> **Q3: what are the specific settings for the different variants in the ablation study?**
>
> **A3:** Baseline-1 directly uses the standard vision state space module (VSSM) to process flattened vision data in our proposed UNet-shaped framework, following the Norm → VSSM → Norm → channel attention layer flow as referenced in [14] for image denoising tasks. Baseline-2 introduces retinex thory \(L=R \ast I\) to Baseline-1, where \(I\) denotes the low light image and \(R\) denotes the reflectance (enhanced image), \(I\) is the illumination map. aims to estimate the illumination map instead of directly predicting the enhanced image, and then restores the enhanced result by \(R=L/I\), as referenced in [13],[43],[45].
>
>
> [C3] Guo X. LIME: A method for low-light image enhancement. In ACM MM. pages 87-91. 2016.
>
> **Q4: Some state-of-the-art works have not been compared (e.g., DiffLL[1]).**
>
> **A4:** As reported in [C4], DiffLL indeed outperforms ours on some benchmark datasets in terms of PSNR and SSIM. As **Reviewer 1rUY** commented, computational complexity and memory cost are also important parts of the LLIE task. We find that more performance and parameters (inference time over 0.1s, model size over 100M) of DiffLL may be required in the case of comparable performance, while our inference time is below 0.1s and model size is only about 2M. Besides, comparing MambaLLIE and the state-of-the-art method on unpaired datasets, our MambaLLIE outperforms DiffLL in terms of perceptual evaluation metrics and visual results. Please refer to Table R4 and Figure R3. In our revised version, we will add citations to this work and discuss it.
>
> [C4] Jiang et al, Low-Light Image Enhancement with Wavelet-based Diffusion Models. Siggraph Asia 2023.
>
> **Q5: The authors should compare more perceptual evaluation metrics.**
>
> **A5:** Please refer to Table R4. We compare two non-reference perceptual metrics, MUSIQ and NIMA, on five unpaired datasets, including LIME, VV, NPE, MEF, and DICM. Experimental evaluations show the superiority of our method over SOTAs with better perceptual evaluation in most comparisons.
>
> **Q6: The color in the last column of Table 2 is incorrectly marked.**
>
> **A6:** Thanks for your careful reading. We will rectify this mistake in the revised version and double-check the color of Table 2.

---

> ### Author Response · Authors · 2024-08-12
>
> ## Dear Reviewer 1bAi
> Thank you for taking the time to review our submission and providing us with constructive comments and a favorable recommendation. We would like to know if our responses adequately addressed your earlier concerns. Additionally, if you have any further concerns or suggestions, please feel free to let us know. We eagerly await your response and look forward to hearing from you. Thank you for your valuable time and consideration!
>
> Best regards,
>
> The authors

---

### Official Review · Reviewer_DUEm · 2024-07-11

**Soundness:** 3
**Presentation:** 3
**Contribution:** 3
**Rating:** 6
**Confidence:** 5

**Summary:**

The authors proposed a Mamba-inspired method for LLIE, which is designed to address some challenges of the existing method, MambaIR. By integrating GLSSB consisting of LESSM and IRSK, the authors could make the model capture a large local receptive field while preserving global understanding natures. The overall pipeline is explained in detail and motivated by sufficient reasoning. MambaLLIE achieved SOTA performance on several benchmarks, significantly outperforming prior works.

**Strengths:**

The overall pipeline is explained in detail and motivated by sufficient reasoning. MambaLLIE achieved SOTA performance on several benchmarks, significantly outperforming prior works. Considering emerging attention on Mamba for low-level vision tasks, the proposed work is timely and can thus attract attention to the community.

**Weaknesses:**

The proposed method is somewhat incremental in that it makes small (although relevant) changes to MambaIR. Since this is not the first work that introduces Mamba for LLIE, a more in-depth analysis and exploration is expected; however, the proposed is a slightly better engineered network structure. The authors are recommended to discuss MambaIR in more detail and clarify technical contributions over MambaIR.

**Questions:**

Although the ERF visualization in Figure 1 is impressive, the authors are expected to provide some numeric results for quantitative performance comparisons.
It is not clear why it is called "retinex-aware selection" in Figure 2(d).
There are several methods dedicated for face detection in low light. It will be interesting to compare face detection performance compared to such methods,

**Limitations:**

The authors adequately addressed the limitations; but some complexity analysis will be helpful for readers to understand the method.

---

> ### Author Rebuttal · Authors · 2024-08-07
>
> **Q1: Technical contributions over MambaIR.**
>
> **A1:** We want to emphasize that our target and contribution are distinct from MambaIR. 1). Our MambaLLIE is specifically designed for the low-light enhancement (LLIE) task, whereas MambaIR is proposed for image restoration, including image super-resolution (SR) and denoising tasks. Our quantitative and qualitative comparisons indicate that MambaIR exhibits suboptimal performance in the LLIE task. 2). Our aim is to leverage global and local context dependency in the state space module, *while MambaIR merely introduces an additional convolution after the vanilla VSSM to restore neighborhood similarity.* Our local-enhanced design essentially introduces the local invariance of convolution on state space model, which integrates the directional scan with local-enhanced into VSSM, and propose a novel state space. By additionally convolving 2D feature into directional scan, our method ensures a closer arrangement of relevant local tokens, enhancing the capture of local dependencies. This technique is depicted in Figure 2 (c). Next, we follow the selection mechanism by parameterizing the VSSM parameters based on the input, allowing the model to filter out irrelevant information and remember relevant information indefinitely, as pointed in [8]. Our proposed LESSM brings new insights into how to aggregate local context features in existing VSSM, addressing this gap and achieving significant improvements in LLIE and even SR tasks. Besides, our IRSK is a novel block designed for the LLIE task, its Retinex-aware prior selects the spatial context regions, primarily inspired by the selective kernel mechanism as referenced in [23]. The ablation study indicates that our IRSK shows better performance than the channel attention layer adopted in MambaIR.
>
> **Q2: About ERF visualization.**
>
> **A2:** Thanks for your positive comment and worthy suggestion. Most existing models, including ours, primarily depend on visualization results to understand the receptive field.  We further statistics the results as follow
>
> | Method| Global Mean Brightness | 20*20 Center Region Mean Brightness |
> |--------|-------------------------|-------------------------------|
> |SNR-Net| 97.59                   | 180.11                        |
> |RetinexFormer| 90.72                   | 187.00                        |
> |MambaIR| 94.08                   | 163.99                        |
> |MambaLLIE| 98.10                   | 198.42                        |
>
>
> **Q3: It is not clear why it is called "retinex-aware selection".**
>
> **A3:** In our framework, we propose a Retinex-aware kernel selective mechanism, where two coupled Retinex-aware priors are used to select the spatial context regions, each with a different receptive field. Hence, we term it as "Retinex-aware selection" in our framework. According to [13], [43], [45], [C3], illumination is a key prior for low-light enhancement, and the maximum values of RGB images can be regarded as a rough illumination prior. We use it in an implicit manner instead of restoring the illumination map or using a single illumination feature for feature guidance.
>
> [C3] Guo X. LIME: A method for low-light image enhancement. In ACM MM. pages 87-91. 2016.
>
> **Q4: Face detection performance.**
>
> **A4:** We investigate the performance of low-light image enhancement methods on face detection in the dark. We use the DARK FACE dataset and randomly sample 300 images for evaluation. The RetinaFace [C4] is used as the face detector and fed with the results of different LLIE methods. We show the results of different methods in Figure R2 and Table R3. In general, MambaLLIE achieves the better mAP score and visual detection result. Please note that the effectiveness of face detection in low-light conditions depends not only on the quality of the enhancement results but also on the specific face detection algorithm employed. In our evaluation, we utilize the pre-trained RetinaFace model to assess the performance of different low-light image enhancement methods to some extent.
>
> [C4] Sefik Serengil and Alper Ozpinar. A Benchmark of Facial Recognition Pipelines and Co-Usability Performances of Modules. In Journal of Information Technologies, volume 17, number 2, pages 95-107, 2024.
>
> **Q5: Complexity Analysis.**
>
> **A5:**  In the domain of low-light image enhancement, the primary challenges encountered include global color degeneration and local noise disruptions.  To address these issues, we have integrated the Mamba architecture into our framework, which excels at capturing the global dependencies of input data to restore. While Mamba-based models have demonstrated commendable performance in tasks such as image super-resolution and segmentation, with linear or near-linear scaling complexity, they tend to yield less than optimal performance in  low-light enhancement task. This limitation is largely attributed to the current state space models’ inadequacy in capturing local dependencies, which is crucial for detailed restoration. Moreover, conventional end-to-end models typically fall short in low-light tasks due to their neglect of illumination priors. To overcome these shortcomings, we introduce two innovative components: the local-enhanced state space module (LESSM) and the implicit retinex-aware selective kernel module (IRSK). These modules are specifically designed to effectively model both local and global degradations within our framework.

---

### Official Review · Reviewer_ahrr · 2024-07-11

**Soundness:** 3
**Presentation:** 3
**Contribution:** 3
**Rating:** 4
**Confidence:** 5

**Summary:**

This paper introduces a low-light image enhancement module (MambaLLIE). This module has a U-shaped structure and each GLSSB block follows the Transformer-based design. LESSM is proposed to capture the spatial long-term dependency. IRSK is proposed to introduce large and selective kernels for enhancing feature-capturing ability. IRSK also introduces illumination guidance by utilising attention mechanisms.

**Strengths:**

1. The usage of the state space model and large selective kernel greatly increases the effective receptive fields. By capturing longer dependency on features, the network can generate better outputs.

2. MambaLLIE achieves state-of-the-art results on low-light image enhancement tasks across various datasets and evaluation methods.

**Weaknesses:**

1. Could this paper state the difference between LESSM and the Vision State Space Module in [14]? It seems these two blocks have similar designs.

2. The core block GLSSB appears to be a combination of [14] and [23]. However, since [14] is proposed for image restoration tasks, this may limit the novelty of this paper.

3. The descriptions of baseline-1 and baseline-2 are not clear.

4. In Section 4.4 Ablation Study, which dataset is used for the ablation study? How about other datasets? If the dataset used in the ablation study is SDSD-indoor, then the baseline-1 result (PSNR 28.87, SSIM 0.865) is quite similar to [14] (PSNR 28.97, SSIM 0.884), which may indicate that the better performance of this paper relies on Retinex-aware guidance rather than a larger receptive field. However, Retinex-aware guidance has already been explored in [3] and "Low-Light Image Enhancement with Multi-stage Residue Quantization and Brightness-aware Attention."

**Questions:**

1. Could this paper provide an analysis or explanation of why MambaLLIE has a larger receptive field in terms of global and local?

**Limitations:**

This paper discusses limitation in Sec 5 and give potential solutions.

---

> ### Author Rebuttal · Authors · 2024-08-07
>
> **Q1: The difference between LESSM and VSSM.**
>
> **A1:** Thanks for your comment, we would like to rebut that the **MAIN** innovation presented in LESSM is the integration of local 2D dependencies into VSSM and formulate a simple yet efficient local-enhanced state space module (LESSM) to aggregate both local and global information. We **formulate a novel state space** in low light image enhancement community as equation 10, while MambaIR merely introduces an additional convolution after the vanilla VSSM to restore neighborhood similarity. Ours local-enhanced design essentially introduces the local invariance of convolution on state space model, which integrates the directional scan with local-enhanced into VSSM.  This technique is depicted in Figure 2 (c). Besides, our local-enhanced term can be regarded as a local consistency constraint for the state space for vision data, guiding the module to learn neighborhood features and thus enhancing robustness. Similar motivation in state space has been discussed in the literature [47], [C1].
>
> [C1] Pfeffermann, Danny, and Richard Tiller. Small-area estimation with state–space models subject to benchmark constraints. Journal of the American Statistical Association 101.476,1387-1397, 2006.
>
> **Q2: The novelty of GLSSB.**
>
> **A2:** We argue that the GLSSB is novel in terms of _*the new exploration of the vision state space model*_ and _*technical improvements for the Retinex-aware low light enhancement task*_.
>
> The low-light enhancement task faces challenges of global color degeneration and local noise disturbance. Our motivation aims to take into account both global and local image restore. Hence, Our GLSSB mainly follow the _*Norm - LESSM - Norm - IRSK*_ flow. Firstly, leveraging the long-range dependency modeling of Mamba [8] with linear complexity, we inherited the global modeling capabilities of VSSM, yet we find that existing vision state space model does not pay enough attention on capturing local dependencies. As noted in **Q1**, our LESSM significantly differs from VSSM [14].
>
> Our IRSK is introduced after LESSM to selectively aggregate Retinex-aware features, which is a structure design based on physical priors for low light enhancement tasks. Specifically, Our IRSK introduces the Retinex-aware prior to select the spatial context regions, primarily inspired by the selective kernel mechanism as referenced in [23]. However, we would like to point that large selective kernel mechanism is **not suitable** for low-level vision tasks. This is because that the padding operation required for expanding a single large depth-wise kernel generates massive invalid information at the edges of feature maps, which inevitably has a negative impact on image restoration. Additionally, the large selective kernel mechanism uses self-aware spatial kernel selection, which limits the network’s ability to focus on physically prior-aware spatial context regions. Given the global modeling ability of our LESSM, we focus on local selective modeling in our IRSK. Our designed Retinex-aware prior only requires a smaller kernel to learn the spatial features, thereby avoiding excessive padding operations and highlighting the Retinex-aware interested regions.
>
> Therefore, we believe our novelty and contribution is on the par with the SOTA methods.
>
> **Q3: The descriptions of baseline-1 and baseline-2.**
>
> **A3:**  Baseline-1 directly uses the standard vision state space module (VSSM) to process flattened vision data in our proposed UNet-shaped framework, following the Norm → VSSM → Norm → channel attention layer flow as referenced in [14] for image denoising tasks. Baseline-2 introduces retinex thory \(L=R / I\) to Baseline-1, where \(I\) denotes the low light image and \(R\) denotes the reflectance (enhanced image), \(I\) is the illumination map. Baseline-2 aims to estimate the illumination map instead of directly predicting the enhanced image, and then restores the enhanced result by \(R=L/I\), as referenced in [13],[43],[45].
>
> **Q4: About Ablation Study.**
>
> **A4:** Sorry for the confusion caused. The dataset used in the ablation study is SDSD-indoor in previous manuscript. As restated in Q3, Baseline-1 has a similar structure to [14], hence its results (PSNR 28.87, SSIM 0.865) are comparable to those of [14] (PSNR 28.97, SSIM 0.884). Compared with Baseline-1, the improved performance when using our model without IRSK benefits from LSEEM. Our results without LESSM indicate the superior performance of IRSK compared to the vanilla channel attention layer [14]. Meanwhile, we supplement the ablation study with tests on two other benchmark datasets to compare the improvements of LESSM and IRSK. Please refer to the Table R2.
>
> **Q5: Why MambaLLIE has a larger receptive field in terms of global and local.**
>
> **A5:** We would like to answer this question from two perspectives: 1) **LESSM**.  Our proposed LESSM addresses this drawback by introducing a local-enhanced term into the VSSM, which retains the global modeling capability and improves the local receptive field. 2) **IRSK**. Our IRSK further enlarges the local receptive field through the Retinex-aware selective mechanism. As pointed out in [23], [C2], selective kernels can capture context information by adaptively adjusting their receptive field sizes according to the input. In our MambaLLIE, our LRSK has a comparatively smaller kernel than SKNet and LSKNet, but combining LESSM with LRSK achieves a larger receptive field in terms of both global and local information.
>
> As shown in Figure 1 (paper), compared with MambaIR, our MambaLLIE indeed has a larger local receptive field, which is an improvement over VSSM. Additionally, ours have a larger receptive field in terms of both global and local information compared to recent Transformer-based methods.
>
> [C2] Xiang Li, Wenhai Wang, Xiaolin Hu, and Jian Yang. Selective Kernel Networks. In CVPR, pages 510-519. IEEE, 2019.

---

> > ### Author Response · Authors · 2024-08-12
> >
> > ## Dear Reviewer ahrr
> > Thank you for taking the time to review our submission and providing us with constructive comments and a favorable recommendation. We would like to know if our responses adequately addressed your earlier concerns. Additionally, if you have any further concerns or suggestions, please feel free to let us know. We eagerly await your response and look forward to hearing from you. Thank you for your valuable time and consideration!
> >
> > Best regards,
> >
> > The authors

---

> > > ### Comment · Reviewer_ahrr · 2024-08-12
> > >
> > > Thank you for your detailed response. My primary concern is that, despite using Mamba to model global information, the improvement over [14] and [3] is limited. Moreover, the proposed method appears to make only minor changes to MambaIR. Furthermore, as shown in Table 4, the enhancement over [14] largely depends on using illumination guidance L_p in Eq. 10 and 12, which is not a novel approach in low-light image enhancement tasks. This raises the question of whether the Mamba architecture has genuinely contributed to low-light image enhancement. Therefore, I maintain my rating.

---

> ### Author Response · Authors · 2024-08-13
>
> Thank you for your valuable time and consideration. We would like to clarify the novelty of our method through the following key points:
>
> 1. **First to Effectively Combine Mamba and Retinex**: Our method is the first to successfully integrate Mamba and Retinex theory and propose MambaLLIE for low-light enhancement task, achieving state-of-the-art results.
>
> 2. **Fundamental Improvement of the State-Space Model in MambaIR**: We introduce a novel state-space model, LESSM (Equation 9), specifically designed for low-light enhancement. This represents a fundamental improvement over the State-Space Model of MambaIR, which merely adds a convolution after the vanilla VSSM to restore neighborhood similarity. In contrast, our LESSM is both simple and effective, as demonstrated by its innovative integration of 2D dependencies into the directional scan (Equation 10), ensuring a closer arrangement of relevant local tokens. Experimental results show that our model improves PSNR by an average of **4.43%** across all datasets compared to MambaIR, while reducing FLOPs by **65.63%** and the parameter count by **46.98%**.
>
> 3. **Retinex-Aware Selective Mechanism (IRSK)**: Our IRSK mechanism expands the local receptive field through a Retinex-aware selective approach by coupling two illumination maps (Equation 12). This allows for *flexible selection* of the region of interest with different kernels, as illustrated in Figure 2(d). In contrast, [3] introduces a relatively fixed illumination feature into the attention mechanism, while *our selective maps could capture the effective interested regions with the normal and reversed feature and illumination maps*. We have discussed the advantages of our selective kernel behavior in **Lines 274-284*.
>
> 4. **Mamba for LLIE**: The low-light enhancement task is challenged by global color degradation and local noise disturbance. This led us to develop a LLIE model that fuses a global model and a local model. Since Mamba is a global-aware estimator with linear or near-linear scaling complexity, we chose it as our global baseline. Additionally, the CNN-based model preserves local 2D dependencies, refining local degradation to enhance LESSM and IRSK. Benchmark and real-world experimental evaluations indicate that our method **outperforms** previous approaches in both qualitative and quantitative comparisons.
>
> We thank you again for your feedback and hope that our explanation can make you better understand our contribution and efforts.
>
> Best regards,
>
> The authors

---

### Author Rebuttal · Authors · 2024-08-07

We thank all reviewers and chairs for your time, constructive comments, and recognition of our work. We appreciate the positive comments on our idea, contributions, and state-of-the-art performance, such as *"novel and interesting,"* *"well illustrated,"* *"attracts attention from the community,"* and *"significantly outperforms prior works."* We believe all concerns have been clearly and directly addressed. Here we also want to summarize a few key clarifications concerning the contributions of our work.

## The novelty of GLSSB

We argue that the GLSSB is novel in terms of _*the new exploration of the vision state space model*_ and _*technical improvements for the Retinex-aware low light enhancement task*_.

Our **Motivation** aims to take into account both global and local image restore. As we know, the low-light enhancement task faces challenges of global color degeneration and local noise disturbance. Global color degeneration suffer from the decrease of illumination, which tends to introduce a global-aware estimator, while the local degradation may contains the large or small regions. Hence, we intuitively introduce Mamba [8] into low light enhancement task, which excels at capturing the global dependencies of input data to restore. Compared with previous CNN and Transformer-based methods, Mamba-based model has exhibit promising performance with **linear or near-linear scaling complexity** in image super-resolution, image segmentation, etc., but still has the **suboptimal results** in low light enhancement task. We find that the existing state space model is limited in modeling local dependencies, leading to a failure in restoring details. Besides, end-to-end models may exhibit suboptimal performance in low-light enhancement tasks because they do not consider illumination priors.

Based on the two aforementioned assumptions, we propose the local-enhanced state space module (LESSM) and implicit retinex-aware selective kernel module (IRSK), which is the modeling of local and global degradation coupled in our framework for low light enhancement task. Therefore, we believe our novelty and contribution is on the par with the SOTA methods.

## The difference between LESSM and VSSM of MambaIR

Our **Main Contribution** lies in enhancing the local 2D dependencies for vision state space modle (VSSM). Specifically, we aim to formulate a simple yet efficient local-enhanced state space module (LESSM) to aggregate both local and global information by introducing additional local-enhanced term into the scan directions for sequence modeling, compared to existing state space methods, our approach requires only the _*small additional number of parameters*_ of VSSM to achieve significant performance improvements.

The **Key insight** between LESSM and vanilla VSSM and its variants: most prior VSSMs use the different directional scan in their sequential state, which is necessary for strong performance on visual tasks. However, it has been empirically observed that CNN seem to work fine for 2D dependency of vision data. Ours local-enhanced design essentially introduces the local invariance into state space model, which can integrate the existing directional scan with our local-enhanced term into VSSM. By additional convolving 2D  information into directional scan, our method ensures _*a closer arrangement of relevant local tokens*_, enhancing the capture of local dependencies. This technique is depicted in Figure 2 (c). Next, we follow the selection mechanism by parameterizing the VSSM parameters based on the input, allows the model to filter out irrelevant information and remember relevant information indefinitely, as pointed in [8]. Besides, our local-enhanced term can be regarded as a local consistency constraint for the state space for vision data, guiding the module to learn neighborhood features and thus enhancing robustness, literatures [47], [C1] likewise added constraints of local term to a state–space model as equation 10 and developed the estimate performance.

As shown in Figure R1 and Table R1, we demonstrate the effectiveness of the proposed LESSM in MambaIR on image restoration tasks, indicating that our approach also has a positive impact on other low-level vision tasks. While the differences between LESSM and the vanilla VSSM may appear relatively small in description, each has its key motivations and shows significant improvements on common benchmarks over the previous models. We believe that simplicity combined with effectiveness is one of the most favored traits in the field of computer vision.

## Summary
As commented in **Reviewers DUEm and 1rUY**, *Considering emerging attention on Mamba for low-level vision tasks, the proposed work is timely and can thus attract attention to the community. Until now, there have been fewer efforts of Mamba dedicated to the low-light image enhancement task.* We posit that our contributions can provide a fresh perspectives on VSSM for low-level vision tasks. Our method achieves the SOTA results on low-light image enhancement, even image super-resolution, across various datasets and evaluation methods. Finally, we are willing to supplement the newly added experiments and analysis in the final manuscript/supplementary material. Also, our code and pre-trained model will be released on our project page.

[C1] Pfeffermann, Danny, and Richard Tiller. Small-area estimation with state–space models subject to benchmark constraints. Journal of the American Statistical Association 101.476,1387-1397, 2006.

---

### Decision · Program_Chairs · 2024-09-25

**Decision:**

Accept (poster)

**Comment:**

The submission received scores of 4, 4, 6, 8. Reviewers recognize that the paper provides some relevant and well-constructed contributions, but there is disagreement about the level of novelty. In particular, reviewers ahrr (elaborated on in the AC-reviewer discussion) and DUEm considered the changes to be mainly incremental compared to MambaIR, while 1bAi considered that the modification follows the general trend of other transformer-based approaches in incprporating local dependencies. In terms of experiments and results, ahrr argued that the improvements are on par with (marginally better) other transformer-based methods, although comparisons were thorough and SOTA had been achieved. Reviewer 1rUY, who gave the score of 8, considered the application of Mamba to the LLE problem to be novel, with other important merits. The authors went into detail in making clarifications in the rebuttal phase.

After taking into account all the material and arguments presented, the AC considers the overall merit of the paper to be sufficient for acceptance. Although the critical comments of the reviewers are valid, and individually the novel components are broadly straightforward, as a whole there is sufficient non-trivial contribution in terms of an extension of Mamba to LLE, modifications that improve the VSSM receptive field (with visualization), and reasonable effectiveness in extending SOTA results.